# On Hedging Properties of Infrastructure Assets during the Pandemic: What We Learn from Global and Emerging Markets?

**Bambang Susantono [1], Gazi Salah Uddin [2,\*], Donghyun Park [3] and Shu Tian [3]**

[1] Knowledge Management and Sustainable Development, Asian Development Bank, Mandaluyong 1550, Philippines; bsusantono@adb.org
[2] Department of Management and Engineering, Linköping University, 581 83 Linköping, Sweden
[3] Economic Research and Regional Cooperation Department, Asian Development Bank, Mandaluyong 1550, Philippines; dpark@adb.org (D.P.); stian@adb.org (S.T.)
[\*] Correspondence: gazi.salah.uddin@liu.se

**Abstract:** Infrastructure investment is essential for economic development for both developed and developing economies. We analyze the short-term return behavior and portfolio characteristics of the global, regional, and selected Asian countries' infrastructure indexes during the pandemic over the sample period 3 July 2018 to 1 July 2021. According to the multivariate Glosten, Jagannathan, and Runkle (GJR) Generalized Autoregressive Conditional Heteroscedasticity (GARCH) with dynamic conditional correlation (DCC) model, infrastructure assets are very heterogeneous depending on the corresponding asset classes. Empirical evidence suggests that infrastructure can be treated as a separate asset sub-class within conventional financial assets. Moreover, we quantify the co-movements between returns on various listed infrastructure indexes and major asset classes, including equity, commodity, currency, and bond index returns. We find that infrastructure assets offer hedging potential against the USD index and USD denominated assets.

**Keywords:** infrastructure asset; equity; crude oil; currency index

## 1. Introduction

The last decade saw an increased interest from institutional and private investors in alternative assets expected to decrease portfolio return variability. This development can be understood in light of the heavy blow dealt with global equity markets by the dot-com bubble and global financial crisis. Infrastructure is one of the investible assets identified by the financial industry as an "alternative" [1]. The demand for infrastructure is considered a new asset class with several attractive and distinctive investment characteristics assigned by the financial industry. In general, infrastructure is viewed as an asset in the transportation, telecommunication, electricity, and water sectors [2]. The investment industry prefers to emphasize infrastructure assets' economic and financial characteristics.

In a broader sense, infrastructure investment is vital for both advanced economies and those in their early stages of development. When roads are built, adequate energy is supplied, and clean water is made available to all, infrastructure may have a revolutionary impact on people's lives and economic potential in developing economies [3]. Maintaining economic development in more developed countries also necessitates meeting demand and investing in new and improved infrastructure.

From the above vein, the need for infrastructure investment between 2016 and 2040 is estimated to be USD 94 trillion, which is 19% greater than existing trends, to satisfy the United Nations' Sustainable Development Goals (SDGs) for electricity and water [3]. According to the Global Infrastructure Outlook, 2017 stylized fact, we may expect Asia to continue to dominate global infrastructure markets in the forthcoming years, given the current trend continues. According to the Global Infrastructure Outlook, investment requirements in America are expected to increase by 47 percent between 2016 and 2040

based on current trends, while investment requirements in Africa are expected to increase by 39 percent. As a result, by 2016–2040, the World, Europe, Oceania, and Asia are forecasting 19%, 16%, 10%, and 10% infrastructural investment deficits, respectively.

Infrastructure investment is also crucial in achieving the SDGs, and the benefits outlined above. Infrastructure is a critical component of SDG 9 (industry, innovation, and infrastructure), and it is seen as having greater long-term advantages for sustainable development [4]. All 17 SDGs and 121 of the 169 targets are influenced by infrastructure, either directly or indirectly [5]. Furthermore, from the sectoral perspective, the two most crucial sectors—electricity and roads—are expected to account for about two-thirds of worldwide investment requirements. Besides meeting the SDGs for the universe, access to drinking water, sanitation, and electricity will necessitate an additional investment of USD 3.5 trillion in global infrastructure by 2030 [3] (Global Infrastructure Outlook, 2017). As a result, infrastructure investments are strongly linked to accomplishing the SDGs.

Despite the above-discussed importance and increased demand for alternative investment avenues, this industry receives little attention when being investigated as an alternative investment vehicle. However, previous research has focused on infrastructure as a distinct asset class. Finkenzeller and Dechant [6], for example, look at the relationship between Australia's property and infrastructure from 1994: Q4 to 2009: Q1. They find that infrastructure can enhance the benefits of diversification. Newell and Peng [7] find that infrastructure is not a sub-class of real estate and that infrastructure provides diversification benefits in the United States over 2000–2006. Bianchi et al. [8] find that the systematic risk factors and industry returns may significantly explain some of the variations in US infrastructure returns from 1927 to 2010. Their findings suggest that infrastructure is a separate asset class.

Alam et al. [9] investigate the impact of the lockdown of the Indian Stock market by using an event study during the period 24 February–17 April 2020 and found that lockdown has a positive effect on stock market performance. Very recently, Uddin et al. [10] investigated the interdependence between the Asian financial markets and the global financial market and found a strong and positive dependence among the analyzed markets due to COVID-19.

Accordingly, Inderst [1] identified business risk, interest rate risk, and political risk as principal risk factors for infrastructure companies and projects. When analyzing infrastructure as an asset class, these are the most critical risk factors to consider. However, infrastructure assets have the following key characteristics: (1) often high entry barriers, monopolistic competition, and strong pricing power, (2) low correlation with the business cycle and other assets, (3) predictable, stable, relatively high free cash flow over a long investment horizon, and inflation-leading, and (4) popular substitute for bond assets in an environment where interest rates are low, and inflation is emerging, such as the current market situation.

The recent COVID-19 pandemic has renewed investors' interest in alternative asset classes such as real estate, commodities, and infrastructure. In particular, infrastructure investments seem to offer interesting diversification opportunities for long-term institutional investors. Some financial analysts suggest that diversifying infrastructure investments across countries has benefits. To close down the existing literature gap, this study analyzes the short-term return behavior and portfolio characteristics of the global, regional, and selected Asian countries' infrastructure indexes during the pandemic. Specially, we would like to investigate the following research questions: (i) How are returns on global, regional, and some Asian countries' infrastructure indexes related to other major asset classes during the pandemic? (ii) Is there any hedging and safe haven potential for infrastructure assets against major asset classes?

To do so, we employ the DCC-GJR-GARCH (1, 1) approach for the following ground. The application of the proposed GJR-GARCH (1, 1) specification received huge scientific attention after the Global Financial and European Debt crises, especially because it is commonly known that financial data incorporate and exhibit asymmetric fat tails and non-

elliptical dependence structural distributions. Utilizing the data from 3 July 2018 to 1 July 2021, the evidence of dynamic conditional correlation unveils that infrastructure assets are very heterogeneous depending on the corresponding asset classes and provide new insights for predicting the investigated markets. The findings also indicate that infrastructure assets offer hedging potential against the USD index and USD denominated assets. Moreover, the role of asymmetric interdependence could be important in employing advanced hedging strategies, portfolio optimization, and new asset allocation techniques for individual and institutional investors.

Nonetheless, our study adds to the existing body of knowledge in a couple of ways. To begin, to the best of the authors' knowledge, the current study, which focuses on the relationship between infrastructure indexes and other financial assets, first gives a significant insight into infrastructure investment as hedging and diversification potentials. Furthermore, we take into account both before and after during the COVID-19 pandemic, allowing investors to build effective portfolios that will protect their investments against pandemic risk. Second, from a methodological standpoint, this study uses the DCC-GJR-GARCH (1, 1) approach to capture the time-varying relationship between infrastructure indexes and other financial assets. Thirdly, our findings suggest that infrastructure assets can be used to hedge against the USD index and USD denominated assets. Finally, investors, policymakers, and governments may use our findings to make a more effective investment, policy, and financing decisions.

The rest of the study is laid out as follows. Section 2 outlines the methodology employed in this study. Section 3 introduces the dataset and summary statistics. The empirical results and discussions are presented in Section 4, and finally, Section 5 concludes the study with some unique policy recommendations.

## 2. GJR-GARCH (1, 1)-DCC Methodology

In this study, we consider the multivariate Glosten, Jagannathan, and Runkle [11] (GJR) Generalized Autoregressive Conditional Heteroscedasticity (GARCH) model with dynamic conditional correlation (DCC) proposed by Engle [12], the GJR-GARCH model. The conditional return on the infrastructure index follows AR (1) process:

$$R_t = \mu + \psi R_{t-1} + \mathcal{E}_t, \qquad \mathcal{E}_t = z_t h_t, \ z_t \sim N(0, 1), \tag{1}$$

where $R_t = [R_{i,t}, \ldots, R_{n,t}]$ is the $(n \times 1)$ vector of the returns on the infrastructure index and an asset class (stock, crude oil, bond, and currency) index under consideration. $\mu$ is the vector of the constant terms, and $\psi$ denotes the coefficient vector of the autoregressive terms. $\mathcal{E}_t = [\mathcal{E}_{i,t}, \ldots, \mathcal{E}_{n,t}]$ represents the vector of standard residuals. In the next step, we estimate the conditional variance (conditional volatility) of the $i$-th element follows from the GJR-GARCH (1, 1) model as follows:

$$h_{i,t}^2 = \omega + \alpha \mathcal{E}_{i-1}^2 + \beta \sigma_{i-1}^2 + Y \mathcal{E}_{i-1}^2 I_{t-1}, \tag{2}$$

where $I_{t-1} = 1$ if $\mathcal{E}_{t-1} < 0$, otherwise $I_{t-1} = 0$. $Y$ is the leverage coefficient term to capture the asymmetric influence. When $Y > 0$, this indicates that the negative shocks impact more than the positive shocks. The parameters $\omega, \alpha, \beta$, and $Y$ in Equation (2) can assure the stationarity of the conditional volatility process only when the conditions $\omega > 0, \alpha, \beta, Y \geq 0$, and $Y + \frac{\alpha + \beta}{2} < 1$ are satisfied.

The diagnostic tests on the standardized squared residuals indicate that our selected GJR-GARCH (1, 1) model with Student-t distribution is correctly specified because the estimated residuals are free from autocorrelation effects. The model also detects the possibility of the existence of the second or higher-order moment of the GJR-GARCH. Finally, the marginal models' residuals adequately capture the return distributions.

We obtain the dynamic correlations between the infrastructure indexes and asset classes (stock, crude oil, bond, and currency) index markets using the Dynamic Conditional Correlation (DCC) model of Engle [12]. Assume that $E_{t-1}[\varepsilon_t] = 0$ and $E_{t-1}[\varepsilon_t \varepsilon_t'] = H_t$,

where $E_t[\cdot]$ is the conditional expectation at the time $t$. Therefore, the conditional variance–covariance matrix ($H_t$) is the product of the variance and correlation matrices and can be defined as the following form:

$$H_t = D_t^{1/2}(DCC)_t D_t^{1/2}, \tag{3}$$

where $DCC_t$ is the $n \times n$ time-varying correlation matrix, while the diagonal matrix of the square roots of the variances is given by $D_t = diag(h_{i,t}, \ldots, h_{n,t})$. Engle [12] models the right-hand side of Equation (4) rather than $H_t$ directly by proposing the following dynamic correlation structure:

$$DCC_t = diag(Q_t)^{-1/2} Q_t diag(Q_t)^{-1/2}, \tag{4}$$

$$Q_t = (1 - a - b)S + a \, diag(Q_{t-1})^{1/2} \hat{\varepsilon}_{i,t-1} \hat{\varepsilon}'_{i,t-1} diag(Q_{t-1})^{1/2} + b Q_{t-1}. \tag{5}$$

This implies that the conditional correlation is dynamically driven by the process of ($Q_t$) where $S$ is the $n \times n$ unconditional covariance matrix for the standardized residuals $\hat{\varepsilon}_{i,t}$, and $a,b$ are non-negative scalars satisfying $a + b < 1$. The resulting model is called DCC-GARCH. The dynamic conditional correlation with different GARCH models is extensively applied in economics and finance-related research. For example, using the Asymmetric Dynamic Conditional Correlation GARCH model, Stoupos and Kiohos [13] find that the business cycle of Denmark, Norway, and Sweden are positively integrated into the European economy. Jiang et al. [14] applied the DCC-GARCH model to study the dynamic relationship between the oil market and China's commodity market.

## 3. Data Construction and Summary Statistics

To capture the return behavior of infrastructure, we consider: three global infrastructure markets, Dow Jones Brookfield Global Infrastructure Total Return Index (DJBGIT Index), MSCI World Infrastructure Gross Total Return USD Index (M2WO0INF Index), and S&P Global Infrastructure Index (SPGTIND Index); two regional infrastructure market indexes, S&P Emerging Markets Infrastructure Net Total Return Index (SPGEIFDN Index) and MSCI AC Asia ex Japan Infrastructure Gross Index (M2ASJINF Index); and three Asian infrastructure markets, S&P BSE Infrastructure Index (SPBSINIP Index), SINUT Index for India, and MSCI Global China Infrastructure Exposure Net Return USD Index (M1CXKSHR Index). These infrastructure resources focus primarily on the economic (transport, energy, utilities, and communications) and social investments (healthcare, education, waste facilities, sports arenas, etc.) and brownfield and greenfield investments.

We consider the several proxy measures of global, regional, and local equity market indexes, including Dow Jones Global Index (W1DOW), MSCI World Index (MXWO), S&P Global 1200 Index (SPGLOB), and S&P500 composite index (S&P500); the regional equity market, including S&P/IFC Emerging Markets (IDRICOTD), MSCI All Country Asia Ex-Japan local (MSELCAXJ); and the Asian equity market, India, China, and Japan. In addition, we consider the major asset classes, including crude oil future contract, currency index, and bond indexes, including FTSE World Government Bond Index (SBWGU) and FTSE Emerging Markets Broad Bond Index (SBEKBBI). The data period covers 3 July 2018 to 1 July 2021. All data are extracted from Bloomberg International. We consider the return series by log price changes between two consecutive periods. We consider 2018 (M7) to 2019 (M12) as the normal period and 2020(M1)–2021(M7) as the pandemic.

The results of descriptive statistics are presented in Table 1. In general, it is determined that all assets' mean returns are positive, although negligible. Specifically, the results reveal that the S&P500 (0.07%) and NCLCS00 (−0.50%) indices exhibit the highest and lowest mean returns across the whole study period, respectively. Conversely, the highest volatility is attached with NCLCS00 (12.43%), while the lowest volatility is associated with bond market (SBEKBBI and SBWGU) (0.32%). Other indices (excluding the USD index) show roughly the same level of volatility (0.38%). (The results of the sub-sample analysis are available on request. To save space, we did not include all sub-sample analyses in the

manuscript.) Except for the SPBSINIP index, all indices show positive and comparable mean returns with nearly similar volatility in the sub-sample 2019. During the sub-sample 2020, however, mean returns are mixed, although most mean returns are positive for all indices (excluding the M2ASJINF and SBWGU indices) throughout 2021. Like the entire sample period, the NCLCS00 index shows the highest volatility during the 2020 sample period, corresponding to the COVID-19 pandemic.

The skewness values for the majority of the indexes are negative across the whole as well as all sub-samples, suggesting that the left side of the distribution's tail is longer or fatter than the right side's tail. Furthermore, for all indices for the same sample periods, the kurtosis values are greater than three, indicating heavier tails than normal distributions. As a result, the returns series distributions for all assets for all the samples are non-normal.

The results from the Jarque–Bera normality test affirms this non-Gaussianity and strongly rejects the null hypothesis of normality at the 1% threshold level, indicating fat tails characterize the distribution. Furthermore, the results of unit root tests, based on ADF and PP tests, are significant, indicating that all series are stationary at levels, while non-stationary at 1st difference based on the KPSS testing approach. Finally, the series suffers from serial correlation and exhibits an ARCH effect according to the Ljung–Box test statistics and Engle's ARCH-LM (5) test. The ARCH-LM test [15] (Engle, 1982) with five lags rejects the null hypothesis of homoscedasticity at the 1% threshold level, signifying the existence of ARCH effects and volatility clustering for all series. This advocates the importance and relevance of the GARCH-type framework to model the stylized facts of the underlying series.

Table 2 shows the results of the correlation matrix among the variables. The findings show strong positive correlations among all the variables (except between USD and crude oil with other indices) during the entire sample period. Moreover, we find that the SBWGU and USD indices have a negative correlation with all other indices in both sub-samples 2019 and 2020, whereas additional variables show a positive relationship between themselves. However, all pairs of the return series (excluding between USD and other indices) have a positive correlation.

The conditional volatility is computed based on the GJR-GARCH (1,1) specifications for all the variables, which is depicted in Figure 1, which indicates that MSCI World Infrastructure Gross Total Return USD Index (M2WO0INF Index) volatility increased between late 2018 and early 2019. Accordingly, the volatility of all the variables skyrockets in the first quarter of 2020. This is not surprising given the COVID-19 pandemic catastrophe that happened at the time. There is abundant evidence that the worldwide pandemic has negatively affected the economic and financial markets. Aside from this volatility, there is little volatility in all variables across the whole study period.

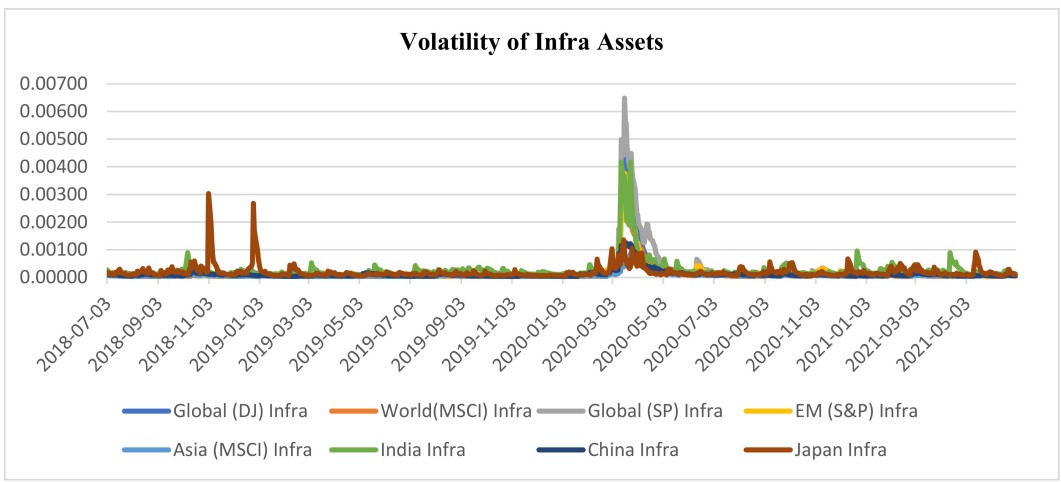

**Figure 1.** Conditional volatility (GJR-GARCH) of infrastructure (global, regional, and selected Asian countries).



**Table 1.** Descriptive statistics.

| Variables | Obs. | Mean (%) | Std. Dev. (%) | Skewness | Kurtosis | Sharpe | Jarque–Bera | ADF(c). Value | ADF(ct). Value | PP Test | KPSS Test | ARCH-LM(5) |
|---|---|---|---|---|---|---|---|---|---|---|---|---|
| DJBGIT | 783 | 0.03 | 1.26 | −2.14 | 31.77 | 0.41 | 27,758.76 *** | −9.24(6) *** | −9.24(6) *** | −31.52 *** | 0.04 | 260.17 *** |
| W1DOW | 783 | 0.04 | 1.13 | −1.67 | 23.78 | 0.62 | 14,537.79 *** | −9.20(6) *** | −9.25(6) *** | −29.86 *** | 0.15 | 241.68 *** |
| M2WO0INF | 783 | 0.03 | 1.08 | −1.78 | 29.44 | 0.46 | 23,354.61 *** | −9.32(6) *** | −9.31(6) *** | −30.20 *** | 0.03 | 260.85 *** |
| MXWO | 783 | 0.05 | 1.20 | −1.49 | 22.97 | 0.6 | 13,379.67 *** | −9.30(6) *** | −9.35(6) *** | −31.39 *** | 0.13 | 258.95 *** |
| SPGTIND | 783 | 0.01 | 1.35 | −2.35 | 35.48 | 0.09 | 35,321.79 *** | −9.07(6) *** | −9.07(6) *** | −30.51 *** | 0.05 | 210.30 *** |
| SPGLOB | 783 | 0.05 | 1.18 | −1.42 | 22.01 | 0.64 | 12,126.05 *** | −9.36(6) *** | −9.41(6) *** | −31.02 *** | 0.13 | 242.68 *** |
| SPGEIFDN | 783 | 0.00 | 1.27 | −1.69 | 18.01 | −0.05 | 7770.20 *** | −11.39(5) *** | −11.38(5) *** | −26.69 *** | 0.07 | 364.23 *** |
| IDRICOTD | 783 | 0.04 | 1.08 | −1.09 | 11.05 | 0.60 | 2285.83 *** | −16.44(1) *** | −16.47(1) *** | −26.46 *** | 0.17 | 257.76 *** |
| M2ASJINF | 783 | 0.00 | 0.87 | −0.54 | 8.97 | −0.08 | 1209.34 *** | −17.73(1) *** | −17.74(1) *** | −29.14 *** | 0.08 | 290.45 *** |
| MSELCAXJ | 783 | 0.03 | 1.03 | −0.56 | 7.47 | 0.51 | 697.15 *** | −17.11(1) *** | −17.13(1) *** | −27.55 *** | 0.18 | 239.80 *** |
| SPBSINIP | 783 | 0.02 | 1.64 | −1.08 | 10.69 | 0.23 | 2095.98 *** | −29.54(0) *** | −29.62(0) *** | −29.60 *** | 0.39 * | 168.32 *** |
| SINUT | 783 | 0.05 | 1.47 | −1.93 | 23.47 | 0.53 | 14,244.53 *** | −11.27(5) *** | −11.35(5) *** | −29.32 *** | 0.25 | 115.78 *** |
| M1CXKSHR | 783 | 0.05 | 1.09 | −1.05 | 11.1 | 0.78 | 2301.41 *** | −12.76(2) *** | −16.39(1) *** | −26.29 *** | 0.16 | 186.89 *** |
| M3CN | 783 | 0.03 | 1.35 | −0.32 | 4.31 | 0.34 | 69.78 *** | −26.48(0) *** | −26.49(0) *** | −26.51 *** | 0.14 | 109.15 *** |
| M2JP0INF | 783 | 0.04 | 1.36 | −0.6 | 9.22 | 0.49 | 1319.36 *** | −31.18(0) *** | −31.17(0) *** | −31.07 *** | 0.06 | 20.71 *** |
| MXJP | 783 | 0.02 | 1.19 | −0.4 | 9.29 | 0.28 | 1319.87 *** | −30.27(0) *** | −30.29(0) *** | −30.20 *** | 0.13 | 66.27 *** |
| S&P500 | 783 | 0.07 | 1.43 | −1.05 | 20.63 | 0.75 | 10,349.92 *** | −9.65(6) *** | −9.69(6) *** | −35.72 *** | 0.10 | 308.12 *** |
| NCLCS00 | 783 | −0.50 | 12.43 | −20.42 | 481.24 | −0.63 | 7,554,754.53 *** | −19.58(1) *** | −19.56(1) *** | −19.32 *** | 0.09 | 24.90 *** |
| SBWGU | 783 | 0.01 | 0.32 | −0.72 | 12.48 | 0.56 | 3019.35 *** | −23.43(0) *** | −23.42(0) *** | −23.38 *** | 0.07 | 219.49 *** |
| SBEKBBI | 783 | 0.02 | 0.32 | −4.49 | 50.19 | 1.22 | 75,694.07 *** | −7.82(6) *** | −7.81(6) *** | −17.89 *** | 0.05 | 244.69 *** |
| USD | 783 | 0.00 | 0.38 | 0.04 | 5.09 | −0.13 | 144.98 *** | −24.67(0) *** | −24.67(0) *** | −24.56 *** | 0.12 | 65.54 *** |

Notes: The table reports the descriptive statistics of the variables used in this research. Std. Dev. denotes standard deviation. '*' and '***' indicate significance levels at 10%, and 1%, respectively.

**Table 2.** Correlation matrix.

| Variables | (1) | (2) | (3) | (4) | (5) | (6) | (7) | (8) | (9) | (10) | (11) | (12) | (13) | (14) | (15) | (16) | (17) | (18) | (19) | (20) | (21) |
|---|---|---|---|---|---|---|---|---|---|---|---|---|---|---|---|---|---|---|---|---|---|
| DJBGIT (1) | 1.00 | | | | | | | | | | | | | | | | | | | | |
| W1DOW (2) | 0.87 | 1.00 | | | | | | | | | | | | | | | | | | | |
| M2WO0INF (3) | 0.94 | 0.85 | 1.00 | | | | | | | | | | | | | | | | | | |
| MXWO (4) | 0.87 | 0.99 | 0.86 | 1.00 | | | | | | | | | | | | | | | | | |
| SPGLOB (5) | 0.87 | 0.98 | 0.85 | 0.98 | 1.00 | | | | | | | | | | | | | | | | |
| SPGTIND (6) | 0.92 | 0.83 | 0.89 | 0.82 | 0.82 | 1.00 | | | | | | | | | | | | | | | |
| IDRICOTD (7) | 0.60 | 0.73 | 0.55 | 0.66 | 0.67 | 0.62 | 1.00 | | | | | | | | | | | | | | |
| SPGEIFDN (8) | 0.73 | 0.77 | 0.65 | 0.72 | 0.74 | 0.74 | 0.80 | 1.00 | | | | | | | | | | | | | |
| M2ASJINF (9) | 0.47 | 0.55 | 0.42 | 0.48 | 0.50 | 0.50 | 0.82 | 0.72 | 1.00 | | | | | | | | | | | | |
| MSELCAXJ (10) | 0.48 | 0.63 | 0.43 | 0.55 | 0.57 | 0.50 | 0.96 | 0.70 | 0.83 | 1.00 | | | | | | | | | | | |
| SINUT (11) | 0.48 | 0.48 | 0.44 | 0.43 | 0.43 | 0.50 | 0.70 | 0.57 | 0.63 | 0.64 | 1.00 | | | | | | | | | | |
| SPBSINIP(12) | 0.39 | 0.39 | 0.35 | 0.36 | 0.36 | 0.44 | 0.52 | 0.46 | 0.46 | 0.46 | 0.75 | 1.00 | | | | | | | | | |
| M1CXKSHR (13) | 0.56 | 0.66 | 0.52 | 0.59 | 0.60 | 0.59 | 0.88 | 0.75 | 0.77 | 0.84 | 0.62 | 0.46 | 1.00 | | | | | | | | |
| M3CN (14) | 0.40 | 0.59 | 0.35 | 0.53 | 0.54 | 0.40 | 0.86 | 0.59 | 0.70 | 0.91 | 0.44 | 0.30 | 0.73 | 1.00 | | | | | | | |
| M2JP0INF (15) | 0.21 | 0.28 | 0.29 | 0.25 | 0.27 | 0.19 | 0.35 | 0.26 | 0.31 | 0.37 | 0.17 | 0.11 | 0.35 | 0.30 | 1.00 | | | | | | |
| MXJP (16) | 0.30 | 0.40 | 0.34 | 0.36 | 0.38 | 0.29 | 0.47 | 0.38 | 0.41 | 0.48 | 0.26 | 0.20 | 0.47 | 0.37 | 0.71 | 1.00 | | | | | |
| S&P500 (17) | 0.82 | 0.95 | 0.81 | 0.97 | 0.95 | 0.75 | 0.54 | 0.64 | 0.37 | 0.44 | 0.33 | 0.29 | 0.46 | 0.45 | 0.16 | 0.24 | 1.00 | | | | |
| SBEKBBI (18) | 0.15 | 0.16 | 0.15 | 0.16 | 0.16 | 0.14 | 0.10 | 0.09 | 0.05 | 0.08 | 0.06 | 0.09 | 0.10 | 0.07 | 0.06 | 0.07 | 0.16 | 1.00 | | | |
| SBWGU (19) | 0.50 | 0.46 | 0.46 | 0.42 | 0.42 | 0.55 | 0.58 | 0.56 | 0.51 | 0.50 | 0.50 | 0.37 | 0.57 | 0.36 | 0.17 | 0.25 | 0.31 | 0.10 | 1.00 | | |
| U$ (20) | 0.01 | −0.05 | 0.03 | −0.07 | −0.08 | 0.05 | 0.04 | 0.01 | 0.05 | 0.01 | 0.07 | 0.01 | 0.06 | −0.01 | −0.04 | −0.12 | −0.14 | −0.01 | 0.39 | 1.00 | |
| NCLCS00 (21) | −0.15 | −0.17 | −0.18 | −0.15 | −0.13 | −0.20 | −0.23 | −0.20 | −0.21 | −0.16 | −0.18 | −0.09 | −0.27 | −0.13 | −0.03 | −0.05 | −0.05 | −0.01 | −0.29 | −0.64 | 1.00 |

Notes: The table presents the correlation matrix among the variables.

## 4. Results and Discussion

Table A1 reports (Appendix A) the AR (1)-GJR-GARCH (1, 1)-DCC model for different infrastructure indices and major financial assets. The parameters of the ARCH term for most of the indices are insignificant. On the other hand, the parameters of the GARCH term are positive and significant in all the cases (except the MXJP index). Furthermore, the GJR (Gamma) parameters are positive and statistically significant for all the indices (except the SPBSINIP, M3CN, MXJP, SBEKBBI, USD, and NCLCS00 indices).

Nonetheless, the results of DCC-GJR-GARCH (1, 1) estimations are robust and reliable since the model has passed several diagnostic tests (Q-statistics and ARCH-LM) and Akaike Information Criteria (AIC) with the Student-t distribution. We observe that most of the Q-statistics and ARCH-LM tests are insignificant, indicating that the data series have no autocorrelation and heteroscedasticity issues.

The results of average DCC parameters indicate that all the infrastructure indices are positively correlated with financial assets except for the USD index, showing no diversification opportunity. Conversely, the USD index has a significant negative correlation with all the infrastructure indices, suggesting that the infrastructure indices exhibit the hedging and safe-haven properties against the USD index. We also notice that the SBWGU and M2JP0INF indices are significantly negatively correlated, highlighting the M2JP0INF index's hedging opportunity against SBWGU.

The results of DCC-GARCH based time-varying correlation between several infrastructure bond indices (e.g., global, regional, and Asian countries) and the different major global financial market indices (i.e., world government bond index, emerging market broad bond index, clean equity index, global commodity index, Eurodollar currency index) are illustrated in Figures 2–4. We observe that all the infrastructure indices and major financial markets (except the SBWGU and USD indices) are positively correlated during the entire study period. This implies that the diversification opportunity does not exist among the variables. The result suggests that the infrastructure indices and other asset classes tend to move together. Hence, it appears that both types of assets are equally impacted by any adverse economic and financial consequences, highlighting that there is no diversification opportunity among the variables. Infrastructure as a distinct asset class lacks sufficient financial theory to support it; instead, it exists as a sub-asset class, or specific sectors, within traditional financial instruments such as equities and bonds [1]. However, our findings contrast with Newell and Peng [7] and Bianchi et al. [8], who claim that infrastructure assets may have portfolio diversification opportunities. One potential reason for why their findings differ from ours is that their studies cover a sample period up to 2010, while our sample period is recent and covers several major crises, such as the COVID-19 pandemic that occurred in late 2019, the oil price war between Russia and Saudi Arabia in early 2020, and so forth. According to recent empirical research, the COVID-19 has impacted practically all sectors and assets of the global economy, resulting in significant financial losses that outweigh previous crises, including the recent 2008 global financial crisis (GFC) [16–18]. The economic and financial crisis caused by COVID-19 is further exacerbated by the recent oil market crisis [16–18]. These might be the likely reasons for the positive association between infrastructure assets and other asset classes.

Interestingly, the USD index displays a persistent negative connection with all infrastructure indices, implying that the infrastructure indices rise when the USD index declines and vice versa. Even if the dollar's value fluctuates, the infrastructure assets are still investable and financing mechanisms to the governments. This demonstrates that the infrastructure indexes act as robust hedge and safe-haven assets for the USD index throughout the sample period. The diversification benefits of infrastructure indices are supported by Bianchi et al. [8]. The strong position of infrastructure investment may be due to the fact that both developed and developing countries are now emphasizing infrastructure development to achieve SDGs. On the other hand, the link between infrastructure indices and the government bond index (SBWGU) varies over time. We also notice that

the SBWGU and infrastructure index, the M2JP0INF index, are significantly negatively correlated, indicating that the M2JP0INF index can be used to hedge against SBWGU.

Existing literature did not explore the infrastructure asset substantially, and such assets remain comparatively unexplored compared to major asset classes. However, our study findings differ from the previous study by [7,8] due to the investigated sample period and economic and pandemic events. They consider the analyzed sample period 2000–2006 [7] and 1927–2010 [8] for United States infrastructure assets and its portfolio diversification with major asset classes. However, we consider the recent COVID-19 pandemic period, which has significantly differed from the previous crisis, and the degree of the contagion's speed among the financial markets [10] due to pandemic related lockdown and several other pandemic related measures.

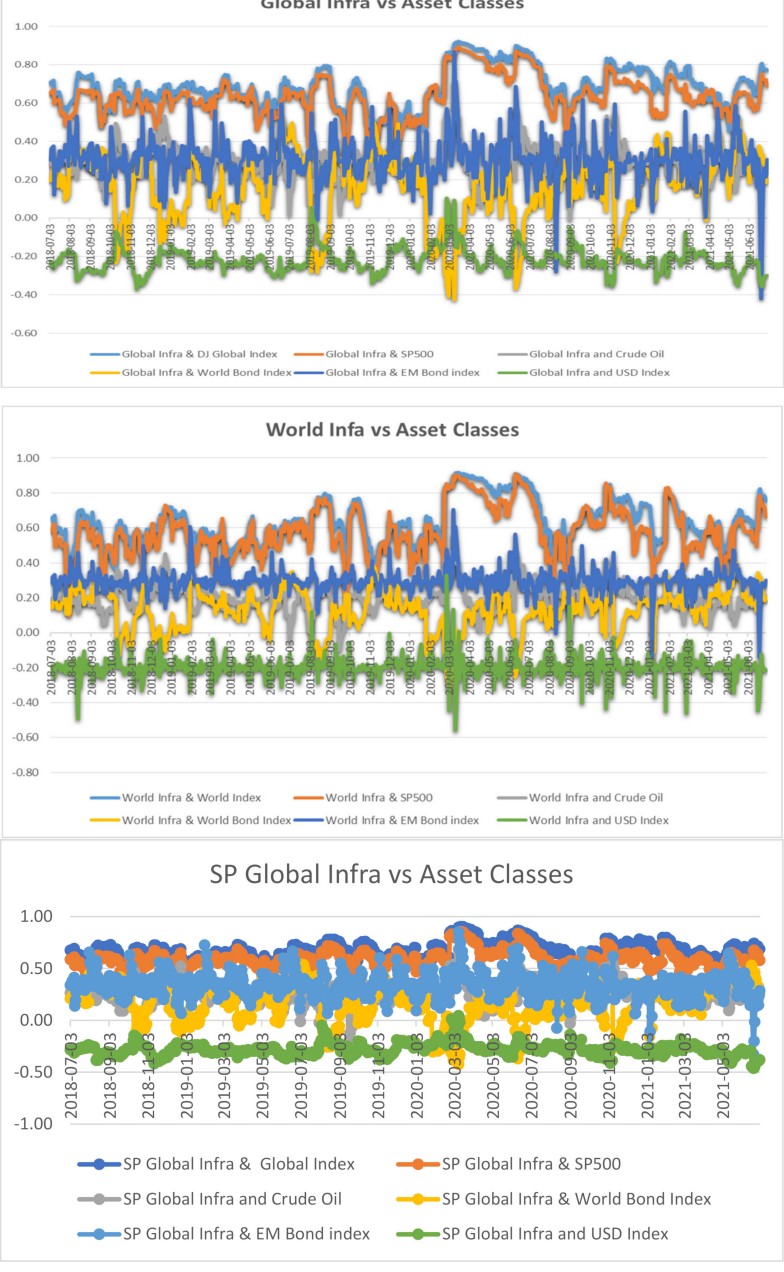

**Figure 2.** DCC-GJR−GRACH between global infrastructure assets and major asset classes. Returns on the Global Infrastructure Index (including Dow Jones, MSCI, and S&P) are persistently and positively related with global equity, bond, and commodity indexes but largely negatively related to the USD index.

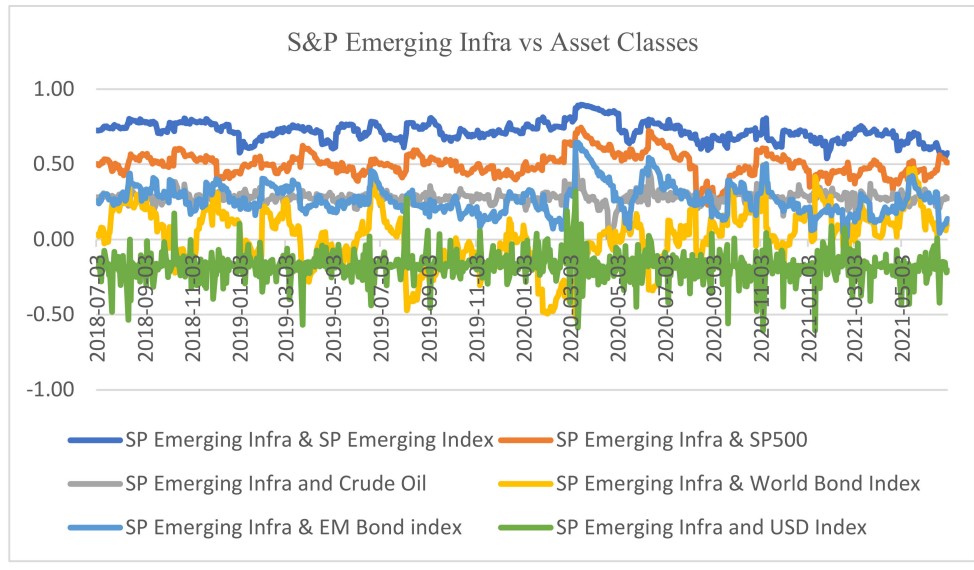

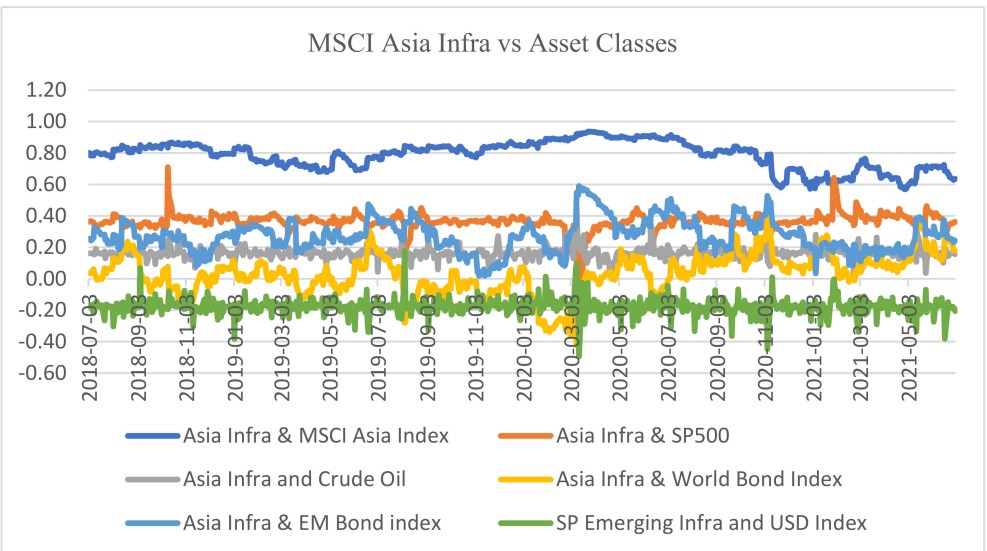

**Figure 3.** DCC-GJR−GARCH between regional (emerging and Asia) infrastructure assets and major asset classes. Returns on regional (emerging and Asia) infrastructure index is persistently and positively related with global equity, bond, and commodity indexes but largely negatively related to the USD index.

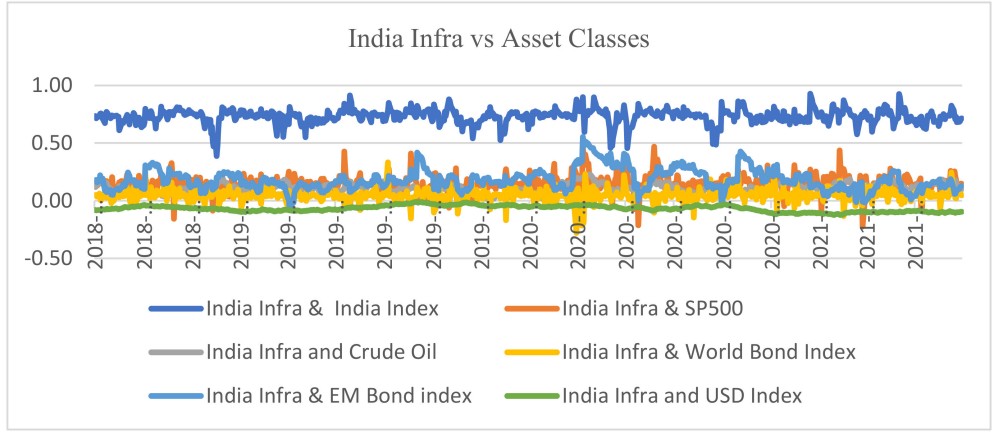

**Figure 4.** *Cont.*

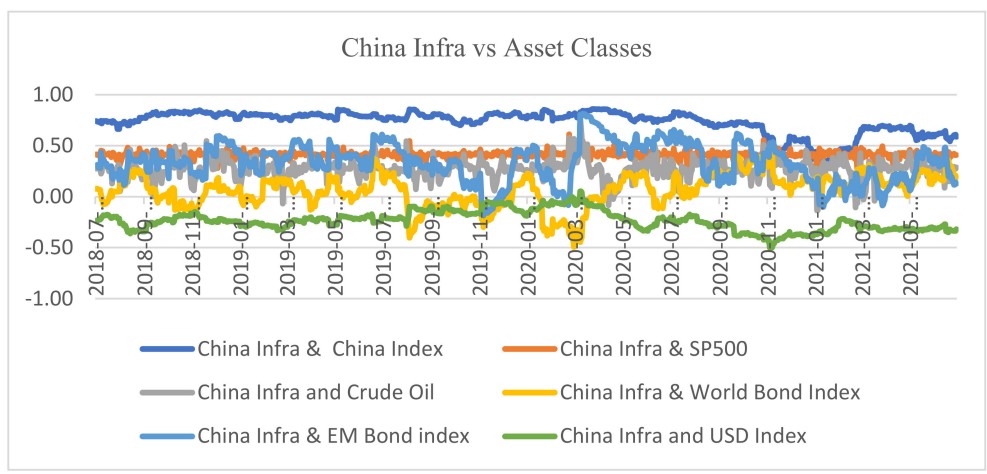

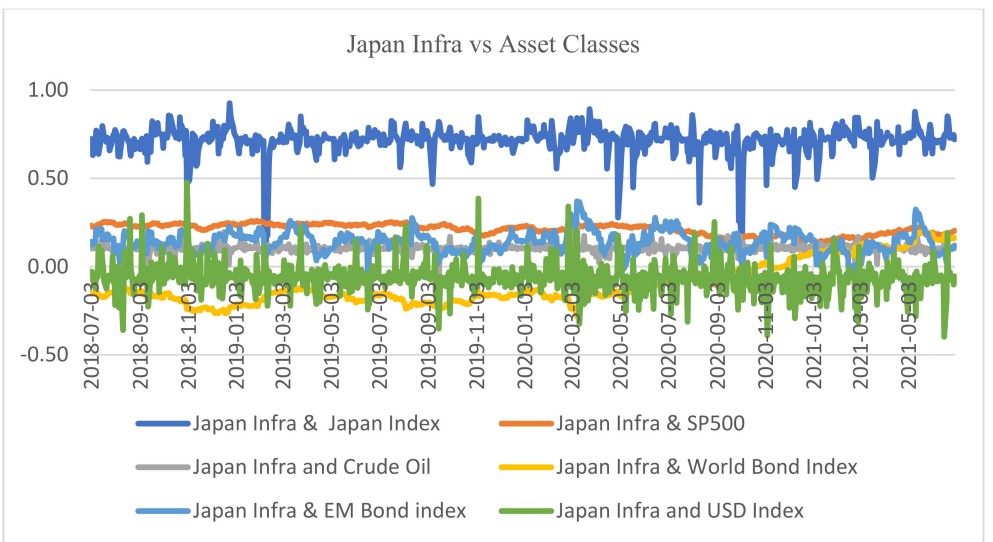

**Figure 4.** DCC-GJR−GARCH between selected Asian infrastructure assets and major asset classes. Returns on selected Asian infrastructure indexes are persistently and positively related to global equity, bond, and commodity indexes but largely negatively related to the USD index.

## 5. Conclusions and Policy Implications

During the COVID-19 pandemic, selected infrastructure indexes' returns for global, developing, Asian, and China have a similar business cycle pattern as major asset classes such as equities and oil prices. We observe that the time-varying correlation between infrastructure indexes and equity, bond, and oil prices are persistently positive for most of the investigated sample period. However, the magnitude of correlation differs over time and corresponding asset classes. Interestingly, infrastructure indexes are persistently negatively related to the USD index, offering potential hedge and safe-haven properties for the USD index and related investments. The negative correlation between the USD and infrastructure indices is consistent with economic intuition.

Global and regional infrastructure indices display a strong co-movement with the equity and commodity markets and a persistent negative correlation with the USD index. In particular, when the USD is strong, returns on infrastructure investment might be lower. The USD index return tends to amplify the hedging effectiveness of the infrastructure market. Governments may utilize the hedging property of infrastructure assets against USD to support their infrastructure investment and related financing decisions. The results have important implications for market efficiency and predictability, particularly in the aftermath of the global financial crisis and the pandemic. Consequently, we attempt to reveal and

further rationalize the financialization mechanism within and across different market channels and its importance for investors, speculators, market makers, and fund managers.

Furthermore, our study suggests that both developed and developing nations should boost their investment in infrastructure development to sustain economic growth. As a critical part of the SDG 9, infrastructure is widely acknowledged as having far-reaching sustainable development benefits. In other words, infrastructure has an influence on all 17 SDGs, comprising 121 out of 169 targets, either explicitly or implicitly (72%). As a result, we recommend that policymakers establish a long-term infrastructure plan with a tiering of policy reforms and investments and plans for closing the infrastructure investment gap to meet the SDGs by 2030. Under this scenario, the continuous deployment of procedures, such as data updates, decisions, knowledge unification, competent supervision, and controlling, may enable adaptable and fit-for-purpose infrastructure development in changing circumstances.

Finally, future studies should include the theoretical model with several measures of uncertainty and its impact on the infrastructure assets using micro-level pandemic country data. In addition, they should carefully identify appropriate instruments to solve the endogeneity problem using advanced econometrics methods.

**Author Contributions:** Conceptualization, D.P.; Data curation, S.T.; Formal analysis, G.S.U.; Investigation, D.P. and S.T.; Methodology, G.S.U.; Project administration, D.P.; Resources, S.T.; Supervision, B.S.; Writing—original draft, G.S.U.; Writing—review & editing, B.S. All authors have read and agreed to the published version of the manuscript.

**Funding:** This research received no external funding.

**Institutional Review Board Statement:** Not applicable.

**Informed Consent Statement:** Not applicable.

**Data Availability Statement:** The data presented in this study are available on Bloomberg.

**Acknowledgments:** We thank four anonymous reviewers for their helpful comments. The second co-author is thankful for the academic financial support provided by the Asian Development Bank. An earlier version of the paper was presented at the Linköping University, Sweden. The authors are thankful to Axel H., Russ Jason N. Lo and Roselyn Regalado for data collection and B. Hasan, M. Yahya, and Ranadeva J. for their helpful comments.

**Conflicts of Interest:** The authors declare no conflict of interest.

# Appendix A. Conditional Volatility and Dynamic Conditional Correlation Model

**Table A1.** GJR-GARCH-DCC Model.

| Variables | DJBGIT | W1DOW | M2WO0INF | MXWO | SPGTIND | SPGLOB | SPGEIFDN | IDRICOTD | M2ASJINF | MSELCAXJ | SPBSINIP | SINUT | M1CXKSHR | M3CN | M2JP0INF | MXJP | S&P500 | SBEKBBI | SBWGU | USD | NCLCS00 |
|---|---|---|---|---|---|---|---|---|---|---|---|---|---|---|---|---|---|---|---|---|---|
| Cst (M) | 0.00 ** (0.00) | 0.00 * (0.00) | 0.00 * (0.00) | 0.00 * (0.00) | 0.00 *** (0.00) | 0.00 * (0.00) | 0.00 (0.00) | 0.00 (0.00) | 0.00 (0.02) | 0.00 *** (0.00) | 0.00 (0.00) | 0.00 * (0.00) | 0.00 * (0.00) | 0.00 (0.00) | 0.00 (0.00) | 0.00 (0.00) | 0.00 * (0.00) | 0.00 * (0.00) | 0.00 *** (0.00) | −0.00 (0.00) | 0.00 * (0.00) |
| AR (1) | 0.08 ** (0.03) | 0.13 * (0.04) | 0.07 ** (0.03) | 0.07 *** (0.03) | 0.06 (0.03) | 0.07 ** (0.03) | 0.13 * (0.04) | 0.17 * (0.04) | −0.02 (0.04) | 0.09 * (0.04) | 0.06 (0.04) | 0.07 *** (0.04) | 0.11 * (0.03) | 0.11 * (0.04) | −0.13 * (0.04) | −0.12 * (0.04) | −0.06 *** (0.04) | 0.29 * (0.04) | 0.10 *** (0.04) | 0.11 * (0.04) | −0.03 (0.04) |
| Cst (V) | 0.03 * (0.01) | 0.02 * (0.01) | 0.02 * (0.01) | 0.03 ** (0.01) | 0.02 ** (0.01) | 0.03 * (0.01) | 0.06 ** (0.02) | 0.05 ** (0.02) | 1.94 ** (0.91) | 0.05 ** (0.02) | 0.28 (0.26) | 0.05 * (0.01) | 0.03 ** (0.01) | 0.14 ** (0.06) | 0.23 * (0.08) | 0.23 (0.23) | 0.04 * (0.01) | 0.22 * (0.80) | 1.02 * (0.28) | 0.20 (0.17) | 1.20 ** (0.54) |
| ARCH (Alpha1) | 0.08 ** (0.04) | 0.06 *** (0.03) | 0.10 ** (0.04) | 0.05 (0.03) | 0.09 ** (0.04) | 0.05 (0.03) | 0.07 (0.04) | 0.03 (0.02) | 0.01 (0.03) | 0.03 (0.02) | 0.14 (0.12) | −0.03 *** (0.01) | 0.02 (0.02) | 0.06 ** (0.03) | 0.20 ** (0.09) | 0.16 (0.11) | 0.06 (0.04) | 0.24 * (0.08) | 0.18 * (0.04) | 0.03 (0.03) | 0.13 (0.08) |
| GARCH (Beta1) | 0.80 * (0.05) | 0.78 * (0.04) | 0.81 * (0.05) | 0.79 * (0.04) | 0.83 * (0.04) | 0.80 * (0.03) | 0.81 * (0.07) | 0.83 * (0.05) | 0.91 * (0.04) | 0.85 * (0.04) | 0.69 * (0.21) | 0.91 * (0.02) | 0.90 * (0.03) | 0.82 * (0.05) | 0.62 * (0.08) | 0.54 (0.28) | 0.78 * (0.04) | 0.69 * (0.07) | 0.74 * (0.05) | 0.94 * (0.03) | 0.70 * (0.05) |
| GJR (Gamma1) | 0.14 ** (0.06) | 0.29 * (0.08) | 0.10 *** (0.05) | 0.29 * (0.08) | 0.12 ** (0.05) | 0.25 * (0.07) | 0.12 ** (0.04) | 0.17 * (0.06) | 0.11 * (0.03) | 0.13 ** (0.05) | 0.11 (0.07) | 0.15 * (0.03) | 0.09 ** (0.04) | 0.08 (0.05) | 0.15 *** (0.09) | 0.34 (0.19) | 0.30 * (0.09) | 0.13 (0.10) | −0.11 ** (0.05) | 0.03 (0.02) | 0.20 (0.14) |
| St (DF) | 7.21 * (1.77) | 6.59 * (1.53) | 5.90 * (1.11) | 5.14 * (0.97) | 6.39 * (1.52) | 5.62 * (1.12) | 7.17 * (1.74) | 9.88 * (3.39) | 5.50 * (0.95) | 5.54 * (1.13) | 5.94 * (1.24) | 4.64 * (0.82) | 5.30 * (1.01) | 10.50 * (3.75) | 6.46 * (1.58) | 5.27 (1.04) | 4.55 * (0.76) | 4.28 * (0.76) | 9.93 * (2.89) | 7.39 * (1.82) | 2.83 * (0.58) |
| Loglik | 2735.51 | 2714.33 | 2814.61 | 2668.37 | 2704.27 | 2668.93 | 2540.6 | 2593.28 | 27.25.3 | 2575.52 | 2215.66 | 2387.06 | 2552.61 | 2305.72 | 2330.13 | 2472.16 | 2538.75 | 3872.48 | 3525.05 | 3298 | 1535.50 |
| Information Criteria | | | | | | | | | | | | | | | | | | | | | |
| AIC | −7.01 | −6.97 | −7.23 | −6.89 | −6.94 | −6.87 | −6.50 | −6.62 | −6.98 | −6.59 | −5.70 | −6.16 | −6.55 | −5.88 | −5.99 | −6.34 | −6.57 | −10.04 | −9.00 | −8.44 | −4.62 |
| Qs2 (20) | 20.88 | 10.13 | 16.37 | 7.40 | 9.18 (5) | 10.61 | 16.19 | 15.35 | 19.74 | 17.40 | 12.15 | 21.91 | 15.40 | 21.06 | 7.30 | 30.05 | 10.42 | 8.91 | 20.26 | 22.38 | 0.07 |
| ARCH (1–10) | 1.73 | 0.68 | 0.57 | 0.49 | 1.05 (5) | 0.76 | 0.33 | 0.38 | 0.36 | 0.49 | 0.55 | 0.62 | 1.10 | 0.85 | 0.33 | 0.88 | 0.84 | 0.61 | 0.85 | 1.39 | 0.01 |
| Estimates of the DCC model | | | | | | | | | | | | | | | | | | | | | |
| DJBGIT | | 0.68 * (0.06) | | | | | | | | | | | | | | | 0.62 * (0.05) | 0.29 * (0.04) | 0.18 * (0.07) | −0.22 * (0.05) | 0.25 * (0.0493) |
| M2WO0INF | | | | 0.62 * (0.07) | | | | | | | | | | | | | 0.56 * (0.07) | 0.28 * (0.04) | 0.13 ** (0.06) | −0.19 * (0.03) | 0.16 * (0.05) |
| SPGTIND | | | | | | 0.65 * (0.04) | | | | | | | | | | | 0.56 * (0.04) | 0.33 * (0.04) | 0.20 * (0.07) | −0.28 * (0.05) | 0.22 * (0.05) |
| SPGEIFDN | | | | | | | | 0.71 * (0.03) | | | | | | | | | 0.49 * (0.04) | 0.25 * (0.0516) | 0.03 * (0.08) | −0.17 * (0.04) | 0.26 * (0.03) |
| M2ASJINF | | | | | | | | | | 0.79 * (0.03) | | | | | | | 0.34 * (0.03) | 0.25 * (0.06) | 0.03 (0.06) | −0.17 * (0.04) | 0.16 * (0.03) |
| SPBSINIP | | | | | | | | | | | | 0.73 * (0.02) | | | | | 0.15 * (0.04) | 0.18 * (0.05) | 0.04 (0.04) | −0.07 *** (0.04) | 0.10 (0.12) |
| M1CXKSHR | | | | | | | | | | | | | | 0.74 * (0.04) | | | 0.38 * (0.03) | 0.31 * (0.08) | 0.07 (0.07) | −0.23 * (0.07) | 0.26 * (0.04) |
| M2JP0INF | | | | | | | | | | | | | | | | 0.70 * (0.02) | 0.19 * (0.03) | 0.14 * (0.04) | −0.15 * (0.04) | −0.06 (0.04) | 0.09 (0.07) |

Notes: The standard errors are in parentheses. '*', '**', and '***' indicate significance levels at 10%, 5%, and 1%, respectively.

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
