# Peer review of "On Hedging Properties of Infrastructure Assets during the Pandemic: What We Learn from Global and Emerging Markets?"

_sustainability, doi:10.3390/su14052987_

Round 1

Reviewer 1 Report

Sustainability-1608191 
On hedging properties of infrastructure assets during the pandemic: What we learn from global and emerging markets?

The purpose 
The authors offer a comparison of behaviour in the infrastructure segment represented by energy and roads on emerging and developed markets. The goal and purpose of the article are well explained. 

Findings 
The paper brings new findings capturing co-movements between returns on major asset classes presented by various listed infrastructure indexes. 
The pandemic influenced the behaviour and brought more caution to the behaviours of investors and countries.
Furthermore, the authors suggest that both developed and developing nations should support their investment in infrastructure development to sustain economic growth. Accepted procedures will enable adaptable and fit-for-purpose infrastructure development in changing circumstances which is necessary for exceptional events like a global pandemic.

Recommendation
The paper has been improved by authors and can now be accepted.  The paper is written in a clear, comprehensive and accessible manner and it is recommended for publishing.

The paper has been improved by authors and can now be accepted.  The paper is written in a clear, comprehensive and accessible manner and it is recommended for publishing.

Reviewer 2 Report

Dear Editor

Thanks for assigning me the task of reviewing the manuscript titled “On hedging properties of infrastructure assets during the pan[1]demic: What we learn from global and emerging markets?”. The study is interesting since it discusses an important issue concerning the global regional and Asian market during the pandemic. However, before it can be considered for publication, the following modifications are required:

  1. In abstract result of the study mentioned, but not mentioned the study period.
  2. Modify the abstract highlight the method.
  3. More in-depth presentation the results required.
  4. Cite the following relevant papers in your study:

https://doi.org/10.13106/jafeb.2020.vol7.no7.131

Author Response

Responses to the Reviewers’ Remarks

Manuscript ID: sustainability-1608191
Type of manuscript: Article
Title: On hedging properties of infrastructure assets during the pandemic:
What we learn from global and emerging markets?
Authors: Bambang Susantono, Gazi Salah Uddin, Donghyun Park, Shu Tian
Decision Received: February 17, 2022

Submission Deadline: Apr 27, 2022

The authors would like to thank the editor for the opportunity to revise our manuscript. We also thank the three anonymous reviewers for providing us with valuable comments and suggestions, which have improved the quality of the paper. We greatly appreciate the opportunity offered to us by the editor and the reviewers to clarify our research contributions. We have examined all the general and specific comments provided by the reviewers and accordingly made the necessary changes to meet their suggestions. We present below our responses to each reviewer’s query.

We hope that these revisions have improved the quality of the present version of the manuscript and will earn the satisfaction of the editor and the reviewer. Our responses to the Reviewers comments are provided in Blue Text

Reviewer 1

Comments and Suggestions for Authors

Dear Editor

Thanks for assigning me the task of reviewing the manuscript titled “On hedging properties of infrastructure assets during the pandemic: What we learn from global and emerging markets?”. The study is interesting since it discusses an important issue concerning the global regional and Asian market during the pandemic. However, before it can be considered for publication, the following modifications are required:

Authors’ response: Thank you for your vote of confidence in our work. We would like to kindly inform you that the paper has been under extensive revisions w.r.t. all concerns/comments requested by the Referee.

  1. In abstract result of the study mentioned, but not mentioned the study period.

Author’s Response: Thank you for highlighting this comment. The suggestion is incorporated in the abstract

  1. Modify the abstract highlight the method.

Author’s Response: Thank you for highlighting this comment. The suggestion is incorporated in the abstract

  1. More in-depth presentation the results required.

Author’s Response: We thank the referee for this helpful comment. The detailed GJR-GARCH (1, 1) model and diagnostic results is presented in the appendix-1. Appendix 1 shows the AR (1)-GJR-GARCH (1, 1)-DCC model for different infrastructure indices and major financial assets. The results of DCC-GJR-GARCH (1, 1) estimations are robust and reliable since the model has passed several diagnostic tests (Q-statistics and ARCH LM) and Akaike Information Criteria (AIC) with the student-t distribution. We observe that most of the Q-statistics and ARCH LM tests are insignificant, indicating that the data series have no autocorrelation and heteroscedasticity issues.

  1. Cite the following relevant papers in your study:

https://doi.org/10.13106/jafeb.2020.vol7.no7.131

Author’s Response: We thank the Reviewer for raising this comment. The suggested article (relevant papers)  is incorporated in the introduction section. The suggestion is addressed as follows:

“Alam et al. (2020) investigate the impact of the lockdown of the Indian Stock market by using an event study during the period 24 February-17 April 2020 and found that lockdown has a positive effect on stock market performance. Very recently, Uddin et al. (2022) investigated the interdependence between the Asian financial markets and the global financial market and found a strong and positive dependence among the analyzed markets due to COVID-19”.

Alam, M. N., Alam, M. S., & Chavali, K. (2020). Stock market response during COVID-19 lockdown period in India: An event study. The Journal of Asian Finance, Economics, and Business7(7), 131-137.

Uddin, G. S., Yahya, M., Goswami, G. G., Lucey, B., & Ahmed, A. (2022). Stock market contagion during the COVID-19 pandemic in emerging economies. International Review of Economics & Finance, 79, 302-309.

Reviewer 3 Report

This study addresses a significant issue on the properties of infrastructure assets from the perspective of sustainability. However, there are still room for further discussions on the return of infrastructure index and that of major financial assets. First, the study identified the co-movement between them, showing no diversification opportunity. The question is whether no hedging properties between them would contribute to the sustainability of infrastructure development and economic growth. Since the author emphasizes the significance of infrastructure development from the SDGs perspective, some discussions are needed in the implication from the estimation outcomes (no diversification opportunity). Second, author argued that this study’ s result is different from Newell and Peng (2008) and Bianchi et al (2014), who claim that infrastructure assets have portfolio diversification opportunities, and that this difference would come from the difference in sample periods. The more discussions seem to be necessary on which sample period should be “normal” (for instance, the recent COVID-pandemic is an exceptional and abnormal period?) or whether the recent period including the pandemic has create some structural change on portfolio diversification opportunities. Third, the conclusion part should contain the limitation and future research issue on this study as in usual academic papers.

Author Response

Responses to the Reviewers’ Remarks

Manuscript ID: sustainability-1608191
Type of manuscript: Article
Title: On hedging properties of infrastructure assets during the pandemic:
What we learn from global and emerging markets?
Authors: Bambang Susantono, Gazi Salah Uddin, Donghyun Park, Shu Tian
Decision Received: February 17, 2022

Submission Deadline: Apr 27, 2022

The authors would like to thank the editor for the opportunity to revise our manuscript. We also thank the three anonymous reviewers for providing us with valuable comments and suggestions, which have improved the quality of the paper. We greatly appreciate the opportunity offered to us by the editor and the reviewers to clarify our research contributions. We have examined all the general and specific comments provided by the reviewers and accordingly made the necessary changes to meet their suggestions. We present below our responses to each reviewer’s query.

We hope that these revisions have improved the quality of the present version of the manuscript and will earn the satisfaction of the editor and the reviewer. Our responses to the Reviewers comments are provided in Blue Text

Reviewer 2

Comments and Suggestions for Authors

This study addresses a significant issue on the properties of infrastructure assets from the perspective of sustainability. However, there are still room for further discussions on the return of infrastructure index and that of major financial assets.

First, the study identified the co-movement between them, showing no diversification opportunity. The question is whether no hedging properties between them would contribute to the sustainability of infrastructure development and economic growth. Since the author emphasizes the significance of infrastructure development from the SDGs perspective, some discussions are needed in the implication from the estimation outcomes (no diversification opportunity).

Response: Thank you for this comment. The result discussion section (Section 4) suggests that all the infrastructure assets provide hedging and diversification opportunities for the USD index and related investments. In a similar vein, MSCI AC Asia ex-Japan Infrastructure Gross Index (M2ASJINF Index) also exhibits the hedging benefit against FTSE World Government Bond Index (SBWGU). That is why we suggest that investors could contribute to the SDGs through investing in infrastructure assets, as these assets provide haven properties against the USD and SBWGU indices. Al the same time, as mentioned in the introduction and conclusion sections, infrastructure investment has a significant influence on all 17 SDGs, comprising 121 out of 169 targets. These are substantially discussed in both Sections 4 and 5. Please see pages 12, 2nd para, and 14-15, 1st and 2nd paras.

Second, author argued that this study’ s result is different from Newell and Peng (2008) and Bianchi et al (2014), who claim that infrastructure assets have portfolio diversification opportunities, and that this difference would come from the difference in sample periods. The more discussions seem to be necessary on which sample period should be “normal” (for instance, the recent COVID-pandemic is an exceptional and abnormal period?) or whether the recent period including the pandemic has create some structural change on portfolio diversification opportunities.

Response: We thank the referee for this helpful comment. Existing literature did not explore the infrastructure asset substantially, and such assets remain comparatively unexplored compared to major asset classes. However, our study findings differ from the previous study by Newell and Peng (2008) and Bianchi et al. (2014) due to the investigated sample period and economic and pandemic events. They consider the analyzed sample period 2000-2006 (Newell and Peng (2008)) and 1927-2010 (Bianchi et al. (2014)) for United States infrastructure assets and its portfolio diversification with major asset classes. But, we consider the recent COVID-10 pandemic period, which has significantly differed from the previous crisis, and the degree of contagion's speed among the financial markets (Uddin et al., 2022) due to pandemic related lockdown and several other pandemics related measures. Our sample period: 3 July 2018 to 1 July 2021. We consider 2018(M7) to 2019(M12) as the normal period and 2020(M1)-2021(M7) as the pandemic.

Third, the conclusion part should contain the limitation and future research issue on this study as in usual academic papers.

Response: Thank you for your comment. We have included both limitations and future research issues in the conclusion section as per your suggestions. Here is the inclusion.

Future studies should include the theoretical model with several measures of uncertainty and its impact on the infrastructure assets using micro-level pandemic country data. And carefully identify appropriate instruments to solve the endogeneity problem using advanced econometrics methods.

We have sent our revised manuscript to a proofreader to ensure the revised manuscript is free of typographical or grammatical errors and acknowledgement and references, the numbering of the sections and pages.

The references are also checked carefully to make sure they are up to date.

Thank you very much for your detailed comments which we believe greatly help improve the content, structure and presentation of our revised manuscript.

We hope that our revised manuscript is publishable in the Sustainability journal.

Reviewer 4 Report

The paper entitled “On hedging properties of infrastructure assets during the pandemic: What we learn from global and emerging markets?” deals with the analysis of short-term return behavior and portfolio characteristics of the global, regional, and selected Asian countries’ infrastructure indexes during the pandemic. The authors have used to the Dynamic Conditional Correlation Model and other method to analyze their data. Τhe authors do not theoretically support their analysis and why they have selected these methods. The results are not tested for problems such as this of double causal relationship. Τhat is why the paper needs a full section of limitations and caveats. Furthermore, the results are not supported and discussed based on the relevant literature. Last the authors do not explain why they haves selected the specific area to be examined. It should be further explained end supported.

Author Response

Responses to the Reviewers’ Remarks

Manuscript ID: sustainability-1608191
Type of manuscript: Article
Title: On hedging properties of infrastructure assets during the pandemic:
What we learn from global and emerging markets?
Authors: Bambang Susantono, Gazi Salah Uddin, Donghyun Park, Shu Tian

Decision Received: February 17, 2022

Submission Deadline: Apr 27, 2022

The authors would like to thank the editor for the opportunity to revise our manuscript. We also thank the three anonymous reviewers for providing us with valuable comments and suggestions, which have improved the quality of the paper. We greatly appreciate the opportunity offered to us by the editor and the reviewers to clarify our research contributions. We have examined all the general and specific comments provided by the reviewers and accordingly made the necessary changes to meet their suggestions. We present below our responses to each reviewer’s query.

We hope that these revisions have improved the quality of the present version of the manuscript and will earn the satisfaction of the editor and the reviewer. Our responses to the Reviewers comments are provided in Blue Text

Reviewer 3

Comments and Suggestions for Authors

The paper entitled “On hedging properties of infrastructure assets during the pandemic: What we learn from global and emerging markets?” deals with the analysis of short-term return behavior and portfolio characteristics of the global, regional, and selected Asian countries’ infrastructure indexes during the pandemic. The authors have used to the Dynamic Conditional Correlation Model and other method to analyze their data.

Τhe authors do not theoretically support their analysis and why they have selected these methods. The results are not tested for problems such as this of double causal relationship. Τhat is why the paper needs a full section of limitations and caveats. Furthermore, the results are not supported and discussed based on the relevant literature. Last the authors do not explain why they haves selected the specific area to be examined. It should be further explained end supported.

Response: Thank you for your comment in our work. We would like to kindly inform you that the paper has been under extensive revisions w.r.t. all concerns/comments requested by the Referee.

Future studies should include the theoretical model with several measures of uncertainty and its impact on the infrastructure assets using micro-level pandemic country data. And carefully identify appropriate instruments to solve the endogeneity problem using advanced econometrics methods.

Existing literature did not explore the infrastructure asset substantially, and such assets remain comparatively unexplored compared to major asset classes. However, our study findings differ from the previous study by Newell and Peng (2008) and Bianchi et al. (2014) due to the investigated sample period and economic and pandemic events. They consider the analyzed sample period 2000-2006 (Newell and Peng (2008)) and 1927-2010 (Bianchi et al. (2014)) for United States infrastructure assets and its portfolio diversification with major asset classes. But, we consider the recent COVID-10 pandemic period, which has significantly differed from the previous crisis, and the degree of contagion's speed among the financial markets (Uddin et al., 2022) due to pandemic related lockdown and several other pandemics related measures. Our sample period: 3 July 2018 to 1 July 2021. We consider 2018(M7) to 2020(M12) as the normal period and 2020-2021 as the pandemic.

Our study provides a comprehensive understanding of infrastructure's return behavior by incorporating the three global infrastructure markets, two regional infrastructure markets, and three Asian infrastructure markets. The recent COVID-19 pandemic has renewed investors' interest in alternative asset classes such as real estate, commodities, and infrastructure. In particular, infrastructure investments seem to offer interesting diversification opportunities for long-term institutional investors.

We have sent our revised manuscript to a proofreader to ensure the revised manuscript is free of typographical or grammatical errors and acknowledgement and references, the numbering of the sections and pages.

The references are also checked carefully to make sure they are up to date.

Thank you very much for your detailed comments which we believe greatly help improve the content, structure and presentation of our revised manuscript.

We hope that our revised manuscript is publishable in the Sustainability journal.

Reviewer 5 Report

In general, it is a good research. However, I believe that some improvements must be done.

1) The authors should add a new section with the title "Literature Review" in order to discover the lacuna of the literature. For instance, I propose two pubished articles about the implementation of the DCC GARCH methodology.

a) Stoupos, N and Kiohos, A. (2019). Scandinavia: Towards the European Monetary Union?. The Quarterly Review of Economics and Finance, 74, 278-291. https://doi.org/10.1016/j.qref.2019.01.006

b) Jiang, Y., Jiang, C., Nie, H., Mo, B. (2019). The time-varying linkages between global oil market and China's commodity sectors: Evidence from DCC-GJR-GARCH analyses. Energy, 166, 577-586. https://doi.org/10.1016/j.energy.2018.10.116

2) The author should discuss the parameters of the DCC-GJR-GARCH, such as volatility persitence, volatility clustering and volatility asymmetry.

3) A professional english proof editing must be done.

I propose a minor revision of this manuscript.

Author Response

Responses to the Reviewers’ Remarks

Manuscript ID: sustainability-1608191
Type of manuscript: Article
Title: On hedging properties of infrastructure assets during the pandemic:
What we learn from global and emerging markets?
Authors: Bambang Susantono, Gazi Salah Uddin, Donghyun Park, Shu Tian
Decision Received: February 17, 2022

Submission Deadline: Apr 27, 2022

The authors would like to thank the editor for the opportunity to revise our manuscript. We also thank the three anonymous reviewers for providing us with valuable comments and suggestions, which have improved the quality of the paper. We greatly appreciate the opportunity offered to us by the editor and the reviewers to clarify our research contributions. We have examined all the general and specific comments provided by the reviewers and accordingly made the necessary changes to meet their suggestions. We present below our responses to each reviewer’s query.

We hope that these revisions have improved the quality of the present version of the manuscript and will earn the satisfaction of the editor and the reviewer. Our responses to the Reviewers comments are provided in Blue Text

Reviewer 4

In general, it is a good research. However, I believe that some improvements must be done.

1) The authors should add a new section with the title "Literature Review" in order to discover the lacuna of the literature. For instance, I propose two published articles about the implementation of the DCC GARCH methodology.

  1. a) Stoupos, N and Kiohos, A. (2019). Scandinavia: Towards the European Monetary Union?. The Quarterly Review of Economics and Finance, 74, 278-291. https://doi.org/10.1016/j.qref.2019.01.006

  1. b) Jiang, Y., Jiang, C., Nie, H., Mo, B. (2019). The time-varying linkages between global oil market and China's commodity sectors: Evidence from DCC-GJR-GARCH analyses. Energy, 166, 577-586. https://doi.org/10.1016/j.energy.2018.10.116

Author’s Response: Thank you for highlighting this comment. The suggested literature is incorporated in the revised version of the manuscript

2) The author should discuss the parameters of the DCC-GJR-GARCH, such as volatility persistence, volatility clustering and volatility asymmetry.

Author’s Response: We thank the referee for this helpful comment. The detailed GJR-GARCH (1, 1) model and diagnostic results is presented in the appendix-1. Appendix 1 shows the AR (1)-GJR-GARCH (1, 1)-DCC model for different infrastructure indices and major financial assets. The results of DCC-GJR-GARCH (1, 1) estimations are robust and reliable since the model has passed several diagnostic tests (Q-statistics and ARCH LM) and Akaike Information Criteria (AIC) with the student-t distribution. The detailed parameters volatility persistence, volatility clustering and volatility asymmetry is discussed at the results and analysis section. We observe that most of the Q-statistics and ARCH LM tests are insignificant, indicating that the data series have no autocorrelation and heteroscedasticity issues.

3) A professional English proof editing must be done.

 We have sent our revised manuscript to a proofreader to ensure the revised manuscript is free of typographical or grammatical errors and acknowledgement and references, the numbering of the sections and pages.

The references are also checked carefully to make sure they are up to date.

Thank you very much for your detailed comments which we believe greatly help improve the content, structure and presentation of our revised manuscript.

I propose a minor revision of this manuscript.

We hope that our revised manuscript is publishable in the Sustainability journal.

Round 2

Reviewer 3 Report

The revision and the explanation are satisfactory, and the revised version deserves the publication. 

Reviewer 4 Report

Τhe authros have rigorously reviewed and effectively responded to all the amendments pointed out in the first revision round. Thus the paper can be accepted in present form.